# Rac GTPases in Hematological Malignancies

**DOI:** 10.3390/ijms19124041

**Published:** 2018-12-14

**Authors:** Valerie Durand-Onaylı, Theresa Haslauer, Andrea Härzschel, Tanja Nicole Hartmann

**Affiliations:** 1Department of Internal Medicine III with Hematology, Medical Oncology, Hemostaseology, Infectious Disease, Rheumatology, Oncologic Center, Salzburg Cancer Research Institute—Laboratory for Immunological and Molecular Cancer Research (SCRI-LIMCR), Paracelsus Medical University, Cancer Cluster Salzburg, 5020 Salzburg, Austria; v.durand-onayli@salk.at (V.D.-O.); th.haslauer@salk.at (T.H.); a.haerzschel@salk.at (A.H.); 2Department of Hematology, Oncology and Stem Cell Transplantation, Faculty of Medicine and Medical Center, University of Freiburg, 79106 Freiburg, Germany

**Keywords:** Rac GTPases, leukemia, lymphoma, microenvironment, cancer, migration, survival, proliferation

## Abstract

Emerging evidence suggests that crosstalk between hematologic tumor cells and the tumor microenvironment contributes to leukemia and lymphoma cell migration, survival, and proliferation. The supportive tumor cell-microenvironment interactions and the resulting cellular processes require adaptations and modulations of the cytoskeleton. The Rac subfamily of the Rho family GTPases includes key regulators of the cytoskeleton, with essential functions in both normal and transformed leukocytes. Rac proteins function downstream of receptor tyrosine kinases, chemokine receptors, and integrins, orchestrating a multitude of signals arising from the microenvironment. As such, it is not surprising that deregulation of Rac expression and activation plays a role in the development and progression of hematological malignancies. In this review, we will give an overview of the specific contribution of the deregulation of Rac GTPases in hematologic malignancies.

## 1. The Rac GTPase Subfamily—Expression, Regulation, and Function

### 1.1. Rac GTPases—Introduction and Mechanisms of Regulation

Hematological malignancies are cancers that affect the bone marrow (BM), the lymph nodes (LN), and the immune system, and encompass acute and chronic leukemias and lymphomas. These malignancies can occur in all developmental stages of the myeloid or lymphoid lineage, such as progenitor cells, stem cells, or specifically differentiated cell types. There is considerable heterogeneity between the entities concerning their dependence on intrinsic oncogenic signaling versus supportive stimuli from the microenvironment. Rac GTPases are critically involved in both tumorigenic factors, and thus have the potential to integrate intrinsic and extrinsic cues, resulting in amplification of pro-survival and proliferation signals [1,2]. The role of the microenvironment in hematological malignancies has long been underestimated, but its importance is now increasingly recognized. Key processes involved in tumorigenesis, i.e., migration, invasion, survival, and proliferation, require reorganization of the cytoskeleton. Rac GTPases orchestrate these rearrangements while acting downstream of receptor tyrosine kinases, chemokine receptors, and integrins, thereby representing integration hubs of numerous microenvironmental signals [3]. However, Rac proteins execute functions far beyond mere control of actin rearrangements, and non-classical roles, e.g., in transcriptional regulation, are increasingly described [4,5,6].

Rho GTPases belong to the Ras superfamily of small GTPases, with over 150 members in mammals. The Rac subfamily of Rho GTPases comprises four members: Rac1, Rac2, Rac3, and RhoG [7,8]. Rac1 is ubiquitously expressed and it is involved in fundamental cellular functions, including embryonic and neuronal development. The importance of Rac1 is reflected by the fact that Rac1-knockout mice are embryonic lethal [9]. Rac2 expression is restricted to cells of hematopoietic origin, whereas Rac3 expression is found predominantly in the brain [8]. It is worth noting, however, that Rac3 was initially identified in a chronic myelogenous leukemia (CML) cell line [10]. RhoG, which shares the lowest sequence similarity with Rac1, is broadly expressed. Rac2-, Rac3-, and RhoG-knockout mice show no apparent altered phenotype [11,12,13]. However, they do have cell-type specific deficiencies, such as macrophages of Rac2^−/−^ mice showing reduced M1 to M2 differentiation potential [13] or RhoG^−/−^ lymphocytes displaying slightly increased antigen receptor cross-linking ability [12].

Like most GTPases, the Rac GTPases are molecular switches that cycle between an inactive state (off-state), in which guanosine diphosphate (GDP) is bound, and an active state (on-state), in which guanosine triphosphate (GTP) is bound. This cycle is highly regulated by different protein families, so-called guanine nucleotide exchange factors (GEFs), GTPase activating proteins (GAPs), and guanine nucleotide dissociation inhibitor (GDIs) [14]. GEFs replace the bound GDP by a GTP, putting Rac into the on-state, essential for downstream effector molecule binding and activation. GAPs in turn enhance the intrinsic GTPase activity, leading to the hydrolyzation of GTP, and termination of Rac activity.

There are GEFs and GAPs contributing to the activation of various Rho family members, but some of them are considered to be (more or less) specific to Rac, such as the GEFs T cell lymphoma invasion and metastasis 1 (Tiam1) [15,16], PIP3-dependent Rac exchanger (P-Rex) [17], and Dock2 [18]. Tiam1 is highly conserved among vertebrates [19] and was first identified in a T lymphoma cell line. In these initially noninvasive cells, Tiam1 expression induced an invasive and metastasizing phenotype [16]. Since today, different groups have confirmed a role for Tiam1 in cell migration and actin cytoskeleton modification in different cancer- and normal cells, such as in gastric cancer or Schwann cells [20,21,22]. Nevertheless, in other cell types, like epithelial cells, Tiam1 as well as Rac1 promoted E-cadherin-dependent cell-cell adhesion and both were shown to restore adhesion of invasive Ras-transformed epithelial Madin-Darby canine kidney (MDCK) cells [23], suggesting cell-type/context dependent output functions of Tiam1. The hematopoietic GEF Vav1 preferentially activates Rac, but as well RhoA and Cdc42, albeit to lesser extents [24].

Rho GEFs comprise approximately 80 members in humans and are subdivided into diffuse B-cell lymphoma (Dbl) and dedicator of cytokinesis (DOCK) family GEFs, according to their structure [25]. The Dbl RhoGEF family consists of around 70 members in humans, including Dbl. Dbl, which was the first GEF for Rho family GTPases to be identified and isolated from a diffuse B cell lymphoma, atcs as an oncogene by its ability to transform fibroblasts [26]. The DOCK family counts 11 members, including DOCK180 (also known as DOCK1), which has been the founding member of this family. Although the two families are structurally unrelated, they have similar functions and substrates.

The first identified GAP was the breakpoint cluster region (Bcr) [25] and until today, 80 further Rho GAPs have been identified. Rho GAPs contain a 150 amino acid conserved domain consisting of nine alpha helices and highly conserved arginine residues in a loop region. These arginine residues form an “arginine-finger”, which is part of the catalytic active site [26,27].

GDIs provide an additional regulatory mechanism by sequestering inactive Rac at the *C*-terminal end in the cytosol, thereby preventing its membrane localization as well as its activation [28].

GEF and GAP activity needs to be tightly spatiotemporally regulated to determine their outcome activity. GEFs and GAPs are typically multidomain proteins, which comprise, in addition to their catalytic domains, mainly protein-protein and lipid interaction domains, allowing their controlled localization and regulation. In addition, some of them even unite both catalytic domains in one protein, such as Bcr, which consequently acts also as a GEF [27]. As a result, GEFs and GAPs do not only regulate Rac through activation and deactivation, respectively, but they also function as scaffolding proteins that bring interaction partners together and enable their proper localization, for example, to the plasma membrane, which is the main site of their action. In addition to the classical regulation of Rac by GEFs, GAPs, and GDIs, another level of regulation is provided by post-translational modifications. These alterations are critical for membrane localization and interaction of Rho GTPases with their GEFs and downstream effectors. Determining this Rho GTPase subcellular location to specific compartments is widely regulated by lipid modifications, such as palmitoylation or prenylation (farnesylation or geranylgeranylation) [29]. In Rac1, palmitoylation targets the translocation to detergent-resistant membrane regions, thereby enhancing its stability. Defects in palmitoylation of Rac1 leads to decreased migration and spreading and the GTPase is less active [30]. Rac GTPases can furthermore be modified by phosphorylation [31], sumoylation [32], and ubiquitylation [33].

### 1.2. Downstream Output of Rac GTPase Activity

Active Rac interacts with downstream effector molecules, like kinases and adaptor proteins, thus determining the corresponding signaling outcome. To attain cell migration and facilitate the required driving forces, actin must polymerize and elongate at the leading edge of the cell, while the actin filaments at the trailing edge must contract. Therefore, GTP-bound Rac acts at the leading edge of a cell, resulting in lamellipodia formation. The Rac targets Wiskott–Aldrich syndrome protein (WASP) and WASP-family verprolin-homologous protein (WAVE) are localized at the front of the lamellipodium, where they activate the Arp2/3 complex and lead to actin polymerization. RhoA is thought to act at the trailing edge of the cell, inducing tail detachment [34,35]. Single cell migration can follow either mesenchymal or amoeboid migration. An elongated cell morphology, dependent on extracellular matrix (ECM) degradation with a slow cell speed, characterizes the mesenchymal form of migration, whereas the amoeboid form of migration is characterized by an amoeboid morphology and a very high cell speed. Rac1 is essential for the mesenchymal migration type, whereas the amoeboid migration type is dependent on RhoA and its downstream effector, Rho coiled-coil kinase (ROCK) [36]. The p21 activated kinase (PAK) family, which is organized in class one (PAK 1-3) and two (PAK 4-6) PAKs, is a major downstream effector protein of Rac GTPases and plays important roles in actin cytoskeleton reorganization and mitogen activated protein kinase (MAPK) signalling. Especially, Rac1 and Rac2 are known for the formation of lamellipodia and membrane ruffling in different cell types, such as fibroblasts [37] or neutrophils [38], respectively. Further, studies on macrophages have revealed that their migration towards M-CSF-1 is Rac dependent, as it was completely abolished when Rac was inhibited [39]. A recent study has further shown that RhoG is involved in growth factor-dependent formation of circular dorsal ruffles (CDRs) in smooth muscle cells [40].

Rac GTPases are further involved in controlling different signal transduction pathways and therefore gene expression [5,6]. Rho, Rac, and Cdc42 activate the c-Jun *N*-terminal kinase (JNK) pathway, a MAPK family pathway. Several MAPK kinase kinases (MAPKKKs) are direct effectors of Rho GTPases, highlighting the link between gene expression and GTPases. Several genes are downstream targets of the MAPK pathway, including genes encoding for cytoskeletal components containing a regulatory serum response element (SRE) in their promotors. GTPases can stimulate the transcription of these genes by interacting with a MAPK, which binds to serum response factors (SRF). SRFs in turn bind the SRE in the promotor and this induces rapid and transient transcription of immediate early genes, stimulating mitogen and growth factors [41,42].

## 2. Rac GTPases in the Hematopoietic System

Rac GTPases have central roles in the hematopoietic system. In hematopoietic stem and progenitor cells (HSPCs) in the BM, Rac GTPases are important for homing and migration by integrating signals from beta-integrins, the chemokine receptor, CXCR4, and the receptor, tyrosine kinase c-kit [43,44]. CXCR4 is the main receptor for CXCL12, a chemokine that is mainly produced by BM endothelial and stromal cells, and functions as an attractant for hematopoietic cells. C-kit is the receptor for stem-cell-factor (SCF), which is also produced by BM stroma cells. The c-kit-SCF axis is critically involved in regulating differentiation and self-renewal of HSPCs [45].

Murine knockout studies have revealed that Rac1 is involved in HSPCs’ homing, engraftment, microenvironmental interaction, and proliferation, whereas Rac2 promotes their migration, adhesion, survival, and retention in niche [2]. Knockout of both Rac GTPases causes a flush out of the stem cells from the BM into the peripheral blood due to decreased adhesion capacity [43,46]. This indicates the intricate role of Rac GTPases in the BM for adhesion and retention of HSCPs. Seminal work from D. A. Williams and his group further demonstrated that Rac1 is also essential for HSPCs homing and engraftment to the BM, as genetic deletion of Rac1 alters the localization of transplanted HSPCs. Instead of homing and engrafting to the medulla cavity and endosteal space of the BM, Rac1^−/−^ HSPCs preferentially infiltrate the spleen. Moreover, microenvironment-dependent hematopoiesis, measured by the frequency of cobblestone area-forming cells in vitro, was strikingly reduced, which indicates that Rac1 is also crucial for microenvironmental interactions of HSPCs [46]. The adhesive capacity of Rac1^−/−^ HSPCs to fibronectin was not affected, and expression of β1 integrins and of CXCR4 was normal, suggesting a different mechanism of altered localization [43]. However, recent data show that overexpression of active Rac1 (Rac1V12) in c-kit^+^ HSPCs has indeed a positive influence on the expression of CD44, VCAM-1, VLA-4 (α4β1 integrin), and CXCR4 [47]. Nevertheless, these data further substantiate the importance of Rac1 in microenvironmental interactions of HSPCs.

Both Rac1 and Rac2 are implicated in T cell activation, division, and migration, as well as in B cell development and B cell receptor (BCR) signaling (for detailed review see [48] and [49], respectively). Though the knowledge of Rho GTPase’s influence on B cell development is limited, Rac2 knockout mice exhibited a decrease of marginal zone B cells due to decreased cell proliferation and Ca2+ influx. Further, a decrease in plasma cells secreting immunoglobulin M (IgM) was detected, leading to an inappropriate humoral immune response. In contrast, Rac1 knockout has almost no effect on B cell development, but the double knockout of Rac1 and 2 shows more severe effects than one knockout alone [50].

Due to their diversity and partial redundancy, it is more difficult to identify specific functions of GEFs and GAPs. Nevertheless, they seem to play an important role in the hematopoietic system, especially in lymphoid cells, since about half of all members of these large families are expressed in these cells in mice [51]. However, there exists expression differences between mice and humans, as, for example, Tiam1 is expressed in human B cells [52], but not in murine ones [51].

As in any tissue, disruption of the normal state and homeostasis of the hematopoietic system by intrinsic or extrinsic events is associated with the development of cancer. Malignant transformation can occur in cells of myeloid or lymphoid origin across all developmental stages, affecting, e.g., stem cells, progenitor cells, or specifically differentiated cell types. Examples are acute lymphoblastic leukemia (ALL) (mostly pre- or pro-B cells), acute myeloid leukemia (AML) (mostly HPSCs), Burkitt lymphoma (mostly germinal center B cells), and chronic lymphocytic leukemia (CLL) (most likely memory B cells). Considering the central role of Rac GTPases in the normal counterparts of these transformed cell types, it is not surprising that Rac pathways are also used by leukemia and lymphoma cells to promote migration and invasion into supportive tissues, proliferation, and survival.

## 3. Altered Expression and Activation of Rac—Phenotypic Observations

Disruption of normal Rac signaling is linked to tumorigenesis and cancer [1,53], which has been extensively documented for solid tumors. However, in many leukemias and lymphomas, altered Rac signaling has been detected as well. In this section, we will give an overview of how and in which leukemias and lymphomas Rac was found to be deregulated, in terms of altered expression, constitutive activity, or mutations (summarized in Table 1).

Aberrant Rac signaling can have different causes, the most obvious one being overexpression resulting in an increased Rac activity. In solid tumors, Rac1 has been found to be overexpressed in prostate [66], breast [67], testicular [68], gastric [69], and lung cancer [70], and Rac3 in breast cancer [71] and colon cancer [72]. In hematological malignancies, overexpression of Rac1 mRNA was observed in multiple myeloma (MM) [65] as well as in mantle cell lymphoma (MCL) [64]. In addition, various MCL cell lines (Jeko-1, Maver-1, Mino, and Z138) show both Rac1 mRNA overexpression and an increased Rac1 activity. Importantly, Rac1 protein overexpression levels correlate with a shorter overall survival in MCL patients, suggesting an active role for Rac1 in the pathophysiology of this lymphoma. Further, Rac1 protein was found to be overexpressed in primary adult AML cells [55], whereas gene profiling of pediatric AML samples revealed that Rac2 is overexpressed in MLL (mixed linage leukemia)-rearranged-bearing AML [56].

A disruption of normal Rac signaling can also result from aberrant upstream input, initiated through GEF overexpression or constitutively active upstream receptors. Overexpression of GEFs, such as Tiam1 in gastric [20], prostate cancer [73], lung adenocarcinoma [74], and in squamous-cell carcinoma of the head and neck (SCCHN) [75], and Vav3 in prostate [76] and gastric cancer [77] may enhance Rac activity. The GEFs, Vav1 and Vav3, have been found to be overexpressed and overactivated—marked by their phosphorylation levels—in primary p190-BCR-ABL^+^ B-ALL patient cells, with Rac2 regulation byVav3 representing the main axis in these leukemic cells [59]. However, in primary murine p190-BCR-ABL^+^ B-ALL cells, elevated levels of active Rac3, but not of Rac1 or of Rac2, were detected [60]. In AML cells that possess the mutated and subsequently constitutive active KITD816V (KITD814V in mice) receptor, Vav1 was constitutively active, resulting in constitutive Rac1 and Rac2 activation [57]. Further, in KITD816V as well as in FLT3ITD^+^ AML cells, focal adhesion kinase (FAK) was constitutively active, resulting in constitutive active Rac1 [58]. Rac1 was also found downstream of the constitutively active tyrosine kinase, NPM-ALK [54], commonly found in anaplastic large-cell lymphoma (ALCL). Vav3 was thereby identified as the major GEF responsible for Rac1 activation. Further, in a human hairy cell leukemia (HCL) cell line, Rac1 was also found to be constitutively active [62]. Of note, in this cell line, RhoA and Cdc42 were also constitutively active. In HCL cells, Rac1 functions downstream of Src and protein kinase C epsilon (PKCε) through Vav GEFs. In this case, Rac1 seems to be directly involved in its own constitutive activity by activating p60Scr, creating a positive feedback loop [63]. Increased Rac1, Rac2, and, to a lesser extent, Rac3 activity were also observed in primary human CD34^+^ peripheral blood cells of CML patients and murine p210-BCR-ABL-expressing cells [61].

It has been long assumed that Rac mutations are not associated with cancer, unlike Ras, which is mutated in about 25% of all tumors [78]. However, recently, next generation sequencing has revealed that Rac is indeed mutated in many tumors and the Rac1 P29S mutation has even been found to be a driver mutation in SCCHN and cutaneous melanoma. So far, using the cBioportal and IntoGen databases, the Rac1 N92K mutant has been identified in MM [53]. However, its specific contribution in this malignancy is not yet known.

Taken together, this shows that Rac GTPases are deregulated in various ways in different hematological malignancies, most frequently affecting Rac1 and Rac2. In the following sections, specific contributions to the various aspects of disease pathophysiology, such as survival and proliferation, are discussed in more detail.

## 4. Contribution of Rac to Leukemogenesis and Lymphomagenesis

### 4.1. Rac1 in Adhesion, Migration, and Engraftment Processes

Malignant hematopoietic cells strongly depend on position signals and effects, which allow spatial proximity to supportive stimuli of the distinct microenvironment [79,80]. Within the tissue, the neoplastic cells destroy the normal hematopoietic structures, dislodge normal cells, and ultimately cause life-threatening dysfunctions. Rac GTPases allow leukemic and lymphomatous cells to massively infiltrate their primary niches, such as the BM and/or the LN, or even infiltrate other tissues, such as the skin, intestine, lungs, and liver. Specifically, Rac1 was found to be involved in the adhesion and migration processes of leukemic and lymphomatous cells (summarized in Table 2).

Of note, several hematological malignancies, including CLL [52] and MM [65], differ significantly from the textbook view of cell motility concerning their usage of Rho and Rac GTPases. Classically, Rac1 is in most cell types associated with protrusion formation at the leading edge of the cell, while RhoA mediates contraction of the trailing edge [36]. In CLL and MM, however, in vitro migration was found to be independent or less dependent on Rac1, respectively [52,65]. Conversely, in these malignancies, the RhoA/ROCK axis seems to be central for migration, with Rac1 acting downstream of RhoA in MM [65]. Rac1 activity in these cells results in adhesion rather than migration, and active Rac1 was even shown to counteract migration in some entities. In Namalwa cells, a cell line derived from Burkitt’s lymphoma, Rac1 activation is prevented through sequestration by association with the formin, FMNL1. Silencing of FMNL1 and subsequent Rac1 activation reduced migration significantly. Blocking of Rac1 by NSC-23766, an inhibitor that is in most cell types Rac1 specific (discussed in more detail in Section 5), in FMNL1-silenced Namalwa cells restored migration, further emphasizing that active Rac1 counteracts migration in this leukemic cell line [85]. Nevertheless, invasion and engraftment processes, which are more complex than mere directional migration, are in many diseases still critically dependent on Rac GTPases. In this case, several signals, including directional cues, but also adhesion molecules and further receptors, need to be integrated, and Rac GTPases represent central hubs of this integration. This is highlighted by the example of MM cells, which migrate independently of Rac, while interactions between Rac1 and integrins are still important for BM homing. A recent in vivo study using NSG mice showed that adhesion of MM cells is highly dependent on α4β1 (VLA-4) integrin and that this process involves Rac1. Talin and Kindlin-3, both cytoskeleton-scaffolding proteins, were identified as functional connections between Rac1 and α4β1 (VLA-4) [84].

In ALL cells, CXCR4 acts together with CD9, a member of the tetraspanin family, in modulating Rac activity [82]. In this study by Arnaud et al., CD9 was shown to colocalize with CXCR4 and increase Rac1 activity via its *C*-terminal tail, in response to CXCL12. NSG mice transplanted with Rac1-depleted REH cells showed longer survival compared to mice that received control REH cells, probably attributable to inefficient engraftment of Rac1-depleted cells. Importantly, CD9 expression also strongly correlates with active Rac1 in primary B-ALL patient blasts and CD9 was recently proposed as a marker for poor prognosis in ALL [86], highlighting an indirect role for Rac1 in the prognosis of ALL.

In AML, a more undifferentiated myeloid neoplasm, Rac1 seems to function in a similar way as it does in normal HSCs. In an AML mouse model in which the leukemia was induced by transplanted murine AE9a (AML1-ETO9a)-transfected c-kit^+^ HSPCs, active Rac1 (Rac1V12) enhances homing and engraftment into the BM. The higher homing and engraftment capacities of Rac1V12-expressing HSPCs are attributable to higher mRNA and protein expression levels of surface molecules, including CD44, VLA-4, and CXCR4 [47]. These findings suggest that active Rac1 is involved in the transcriptional regulation and/or surface expression of these critical homing, migration, and engraftment factors, highlighting the different levels on which Rac GTPases orchestrate the homing and engraftment behavior of malignant cells.

Similarly, Rac1 and Tiam1 seem to be crucial in the infiltration of adult T cell leukemia/lymphoma (ATL) [83]. In different ATL cell lines (MT-2 and ATL-3I), Rac1 was found to be activated by Tiam1, leading to the formation of lamellipodia and adhesion to stromal cells in in vitro cultures. Necessary for this process is the immunoglobulin-like cell adhesion molecule, CADM1. CADM1 interacts with Tiam1 via its PDZ domain and colocalizes in these cell lines at the sites of lamellipodia. The colocalization of CADM1 and Tiam1 was also found in the lymph nodes of a proportion of the ATL patients examined in this study.

In the case of ALK-rearranged anaplastic large cell lymphoma (ALCL), invasiveness and dissemination of the tumor cells are dependent on the interaction of Rac signals with the oncogenic kinase, NPM-ALK. In ALK^+^ ALCL cell lines as well as in NPM-ALK-transformed fibroblasts, Rac1 was a downstream target of NPM-ALK, resulting in its constitutive activation [54]. Independent studies have showed that Rac1 activation downstream of NPM-ALK could be accomplished by the GEFs, Vav3 [54] or Tiam1 [81]. In both studies, the invasiveness of ALK^+^ cells was Rac1-dependent. Activated/phosphorylated Vav3 was also detected in ALK^+^ ALCL LN patient samples, but not in ALK^−^ ALCL LN samples or healthy peripheral blood lymphocytes, underscoring the importance of this finding [54]. Activation of Rac1 by Tiam1 was dependent on the lipid messenger, phosphatidylinositol 5-phosphate (PtdIns5P). PtdIns5P formed complexes with Tiam1 and Rac1, and active Rac1 was localized in ruffles and small vesicles of lamellipodia. Importantly, both studies were conducted with the same cell lines, showing that Rac1 activation encompasses several activation mechanisms in the same system. Of note, Tiam1 recruitment by PtdIns5P and subsequent Rac1 activation seems to be a general mechanism as it has been found not only in NPM-ALK^+^ cells, but in different pathophysiological models, such as ectopic expression of bacterial IpgD, therefore, being relevant particularly in pathologies associated with elevated PtdIns5P [81].

Additional studies based on different transgenic mouse models have confirmed that Rac1 is essential for dissemination and invasion of NPM-ALK^+^ tumor cells [87]. Treatment of mice with NSC-23766, whereby treatment was started at the timepoint of disease onset, prevented the infiltration of spleen, liver, and lungs and reduced the dissemination of NPM-ALK^+^ tumor cells. Analyses of the activation status of various signaling molecules suggested that p38, Erk1/2, and Akt are downstream of Rac1 and are likewise involved in this process. These results highlight the role of Rac1 and its upstream GEFs in the invasion and dissemination of ALK^+^ ALCL, and the targeting of Rac1 is promising in these tumors. ALK-rearrangements are also common in other cancer entities, such as, for instance, in non-small cell lung cancer. Whether Rac1 can serve as a downstream target of ALK in these cancers as well, and whether and how it contributes to invasion of tumor cells in these cancers remains to be elucidated.

Taken together, these observations clearly show that Rac1 plays an important role in adhesion, migration, engraftment, and dissemination of leukemia/lymphoma cells. There are many similarities to normal hematopoietic cells, especially in complex processes, like engraftment. However, some leukemia/lymphoma cells have found a way to modulate Rac1 activity by upstream regulators, such as CD9 in B-ALL and CADM1 in ATL.

### 4.2. Rac in Survival and Chemoresistance Mechanisms

Evading apoptosis, the programmed cell death, is a hallmark of cancer cells. Central apoptotic signaling pathways are often disrupted in hematologic malignancies, shifting the normal balance towards a favorable pro-survival state [88]. This gains further relevance in the context of therapy resistance [80]. Rac GTPases were found to be critically involved in regulating and modulating these survival pathways and contribute therein to chemoresistance mechanisms (summarized in Table 3) in various hematological malignancies.

There is growing evidence that Rac promotes the survival of leukemic and lymphomatous cells through regulation of the expression of Bcl-2 family members. In a murine MA9 (MLL-AF9) AML model, Rac2 was shown to positively regulate the expression of anti-apoptotic Bcl-xL [91]. Mixed linage leukemia (MLL) gene-rearrangements are associated with intermediate to poor prognosis in AML and occur through translocations. One of the most frequent translocations, t(9;11), results in the fusion protein, MLL-AF9 [97]. In vivo knockout studies targeting either Rac1 or Rac2 revealed that only the loss of Rac2 delays or even prevents the onset of MA9-induced AML [91]. Furthermore, in Rac1 knockout mice that develop AML, Rac2 expression was increased, underscoring the critical role for Rac2 in AML initiation in this model. However, in already fully transformed murine MA9 leukemia cells, depletion of either Rac1 or Rac2 reduced the in vitro survival through enhanced apoptosis [91]. The increased apoptosis in the Rac2-deficient cells was due to reduced expression of anti-apoptotic Bcl-2 and Bcl-xL. Blocking of active Rac in human MA9^+^ cell lines likewise resulted in decreased Bcl-xL expression levels, further suggesting Rac-dependent Bcl-xL regulation [91]. Of note, a very recent study by Nimmagadda and colleagues suggests a role for Rac1 in the survival of MLL- rearranged AML cells. In this study, Rac1 was found to be downstream of BTK and inhibition of either BTK or Rac1 resulted in cell death [90], indicating a Rac1-dependent survival mechanism. Similarly to MA9 AML, in murine cells expressing KITD814V, a mutation found in about 40% of AML cases and in about 90% of systemic mastocytosis (SM) cases, and treatment with EHop-016, an inhibitor that targets Rac1 and Rac2 (discussed in more detail in Section 5.1), resulted in reduced proliferation and cell viability due to enhanced apoptosis through repression of the Bcl-2 family member, Bad [57]. In AML cells harboring FLT3 internal tandem duplication (FLT3/ITD) mutation, a genetic aberration found in about 30 % of all AML patients, constitutively active Rac1, in concert with PAK1, was shown to bind directly to the signal transducer and activator of transcription 5 (STAT5), inducing STAT5 phosphorylation at Y694. This complex of active-Rac-active-Stat5 translocates into the nucleus and results in the transcription of c-myc and Bcl-xL, likely contributing to the survival of these cells [58]. Rac1 has also been implicated in the survival and lymphomagenesis onset of NPM-ALK^+^ ALCL [89]. In this study, NPM-ALK transgenic mice exhibiting T cell specific Rac1 or Cdc42 single or double deletions were used. Both single deletions delayed the onset of lymphomagenesis and increased survival of the mice. Double deletion of Cdc42 and Rac1 even completely abrogated the development of lymphoma up to 28 weeks. At this point, the mice died due to multiorgan failure of thus far unknown reasons. Histological analysis of Rac1^−/−^ and Cdc42^−/−^ single knockout tumors revealed a comparable proliferation, but strikingly higher apoptotic rate compared to control tumors. The knockout tumors further displayed increased proapoptotic Bid expression. By overexpression of Bcl-2, upregulation of Bid and caspase-3 can be prevented, and apoptosis is consequently suppressed. In p190-BCR-ABL^+^ B-ALL cells, the GEF, Vav3, was observed to be pivotal for the survival of these cells through repression of the proapoptotic proteins, Bad, Bax, Bak, and Bik [59]. Depletion of Vav3 in murine p190-BCR-ABL-transduced B cell progenitors abrogated their enhanced survival capacities in vitro, whereas depletion of Vav1/2 results in even lower apoptosis. This unexpected observation in the Vav1/2 knockout cells was explained by a compensation by Vav3; Vav3 is upregulated in the Vav1/2 knockouts, highlighting its pro-survival role. In addition, Vav3 is in this system required for full activation of total Rac, Rac2 as well as Cdc42 and RhoA, with Rac2 being most strongly bound and activated by Vav3 [59].

Besides their direct or indirect involvement in transcriptional regulation, Rac GTPases also promote pro-survival signals through modulation or direct binding of Bcl-2 family members or through direct interaction with mitochondrial proteins. In a Burkitt’s lymphoma cell line (BL-41), Rac1 was found to induce the phosphorylation of Bad at S75 (corresponds to murine Ser112), which keeps Bad inactive and prevents its pro-apoptotic properties. Phosphorylation at this site was mediated by cAMP-dependent protein kinase (PKA) in response to active Rac1 whereas Akt was not involved [95]. In different lymphoblastic cell lines (CEM, Jurkat, and Raji) Rac1 was capable of binding directly to Bcl-2, thereby stabilizing the anti-apoptotic properties of Bcl-2 and contributing to chemoresistance towards vincristine and etoposide (VP-16) [93]. The Rac1-Bcl-2 complex is preferentially located at mitochondria and promotes an increase in intracellular superoxide via involvement of STAT3 phosphorylation [94] that inhibits apoptosis. Apoptosis of lymphoblastic cell lines induced by etoposide and vincristine was only efficient in combination with BH3 peptides disrupting the Rac1-Bcl-2 complex. Notably, this interaction was also found in B cell lymphoma patient samples whereas it was not observed in noncancerous tissue. In the case of chronic myeloid leukemia (CML), Rac2 directly interacted with mitochondrial proteins, thereby ensuring the survival of these cells [96]. Specifically, the loss of Rac2, but not of Rac1, induced apoptosis of CD34^+^, BCR-ABL-transduced cord blood (CB) cells, and was associated with decreased cell growth and cell cycle arrest. The increased apoptosis in Rac2^−/−^ cells in this model was due to a reduced mitochondrial membrane potential and an altered mitochondrial ultrastructure, resulting in mitochondrial dysfunction. Tandem mass spectrometry of a CML cell line (K562) revealed that Rac1 interacts preferentially with cytoskeleton and adherents junction associated proteins, such as vinculin, emerin, and junction plakoglobin. Rac2, in contrast, interacts predominantly with mitochondrial proteins, such as SAM50 and Metaxin-1, thereby promoting mitochondrial integrity. Depletion of SAM50, which is a mitochondrial outer membrane protein, results likewise in increased apoptosis. Consequently, SAM50 was recently proposed as an important mediator of survival in BCR-ABL^+^ leukemic HSPCs [98].

Another pathway of apoptosis induction is initiated in response to genomic instability and DNA damage. Rac1 was found to be involved in this pathway in monocytic MLL-rearranged AML cells, where its inhibition resulted in DNA double strand breaks and consequent caspase activation, as well as downregulation of the pro-survival factors, BIRC4, BIRC5, and p-Akt [99]. Rac1 is thus involved in maintaining genomic integrity in these cells.

Interaction of Rac1 with another central mediator of apoptosis in response to DNA damage, p53, was revealed in p53 deficient B and T cell lymphoma cell lines (BL-41 cells and J3D cells, respectively) [100]. Deletions or mutations of the p53 gene are commonly found in many leukemias and lymphomas and are a marker of poor prognosis and poor treatment outcome [101]. P53-deficient cell lines exhibited increased Rac1 activity, which was associated with enhanced survival and proliferation (discussed in more detail in Section 4.3). Blocking of Rac1 by different approaches induced apoptosis, accompanied by cleaved caspase-3 and cytochrome c release.

We have identified Rac1 and its GEF Tiam1 to be critical in the chemoresistance of CLL cells [52]. Activated CLL cells can develop a relevant resistance against fludarabine when co-cultured with stromal cells and activated T cells. Signals from this protective microenvironment trigger Tiam1 expression in CLL cells and Rac1 is activated, promoting CLL cell chemoresistance and proliferation (discussed in detail in Section 4.3). Importantly, pharmacological inhibition of Rac1 resensitized CLL cells towards fludarabine [52]. Moreover, in CLL, resistance towards ibrutinib is associated with mutations of phospholipase Cγ2 (PLCγ2), which is part of the BCR signalosome [102]. Ibrutinib is a small molecule inhibitor of BTK that effectively prevents CLL proliferation [103]. Rac2 was found to activate constitutively two known mutants of PLCγ2 occurring in CLL, namely R665W and L845F [104]. These findings suggest that Rac2 can activate the BCR signalosome independently of BTK and thereby likely making an important contribution to the survival and proliferation of CLL cells bearing these mutations.

A role of Rac1 in treatment resistance was further found in an AML cell line (KG-1a). VP-16 treatment of these cells, in combination with Rac1 inhibition, resulted in significantly higher apoptosis rates compared to VP-16 only treatment, implicating Rac1 in the chemoresistance of these cells [92]. Gene expression profiling of pediatric AML and ALL patient samples revealed a potential role for RhoG in the chemoresistance towards VP-16 [105].

### 4.3. Rac in Proliferation Processes

Proliferation requires the entry and progression of the cell cycle, which includes dynamic changes and reorganizations of the actin cytoskeleton. Thus, it is not surprising that there are many reports demonstrating a role for Rac in the regulation of the cell cycle [106,107,108]. For example, Rac1, in concert with Tiam1 and members of the PAK family, has been shown to balance the force of the mitotic spindle of MDCK II cells [109]. In the case of hematological malignancies, Rac GTPases have been implicated in uncontrolled proliferation, triggered by oncogenic transformation or/and microenvironmental interdependencies (summarized in Table 4).

In addition to our observation of Tiam1/Rac1 involvement in the chemoresistance of CLL cells towards fludarabine (see previous paragraph), we propose a direct role of Tiam1/Rac1 in cell cycle progression of CLL [52]. The proliferation process of CLL cells depends strongly on supportive stimuli of the microenvironment, which can be mimicked in vitro by co-culturing CLL cells with stromal cells, CD154 (CD40 Ligand, CD40L) stimulation, and/or activated T cells [112]. In comparison to resting, non-proliferating CLL cells, Rac1 was found to be expressed and activated more in dividing CLL cells. Notably, Tiam1 expression, normally low or almost absent in resting CLL cells, was massively induced in CLL cells that received proliferative stimulation, and was identified as the main Rac1 GEF in this process [52]. Proliferation was thereby accompanied by a Tiam1/Rac1-dependent induction of c-myc [52], a transcription factor involved in cell cycle progression and proliferation, which is frequently deregulated in B cell tumors [113]. Later, Kipps and colleagues observed that the Rac1 activity in CD40L-induced proliferating CLL cells can be further enhanced by exogenous Wnt5a [110]. They found that Wnt5a induces the heterooligomerization of ROR1 and ROR2, both orphan receptors that are highly expressed in embryogenesis and are found on primary CLL cells [114]. This heterooligomerization leads to the recruitment and colocalization of the GEFs, ARHGEF2, ARHGEF6, and DOCK2 [111], resulting in downstream Rac1 and Rac2 activation. Of importance, Wnt5a in combination with CD40L is also able to activate Rac1 in CLL cells treated with ibrutinib, resulting in enhanced proliferation. In addition, Rac1 was found to be active in freshly isolated CLL cells from patients that were treated with ibrutinib [115]. As some patients display progressive disease under ibrutinib treatment, this suggests alternative proliferation signaling pathways in these patients, likely involving the ROR1/Rac1 axis. Indeed, an anti-ROR1 monoclonal antibody, namely cirmtuzumab (UC-961), is being tested at the moment alone or in combination with ibrutinib in several phase 1b/2 studies for the treatment of B cell lymphoid malignancies, including CLL [116].

As described before, increased Rac1 activity was observed in p53 deficient B and T cell lymphoma cell lines (BL-41 cells and J3D cells, respectively) [100]. P53 functions as a key tumor suppressor in most cell types and germ line deletion of p53 in mice is sufficient to induce lymphoma development, emphasizing its critical role in lymphomagenesis [117]. Xenograft studies of BL-41 cells in immunodeficient NOG mice revealed that silencing of Rac1 markedly suppressed tumor growth compared to Rac1-proficient BL-41 cells. In addition, the few tumors that developed from shRac1 lymphoma cells were found to express Rac1, suggesting that incomplete silencing of Rac1 facilitated the proliferation of these cells. Similarly, transplantation of murine p53^−/−^ BM cells with inducible Rac1 knockout into BOYJ recipients showed a significant delay in the de novo onset of lymphoma and increased survival of the mice in the absence of Rac1. Importantly, this was not observed for Rac2 deletion, which displays no difference in survival compared to the control. In vitro inhibition of Rac1 is also able to reduce the hyperproliferation of p53-deficient lymphoma cells [100]. As p53 dysfunctionality is associated with poor prognosis and poor responsiveness to conventional treatment options, Rac1 might be a promising drug target for these patients.

Abnormalities of another important tumor suppressor, phosphatase and tensin homolog (PTEN), a phosphatase that counteracts phosphoinositide 3-kinase (PI3K) signaling, are frequently observed in hematopoietic malignancies [118]. Mice lacking PTEN expression in the BM show an increased amount of myeloid and lymphoid cells and develop myeloproliferative neoplasms (MPN) [119]. A recent study discovered a central role of Rac1/2 in MPN development in the setting of PTEN-loss. In this study, a positive feedback loop of the PI3K isoform, p110β, and Rac1/2 was found that promotes in vivo expansion as well as in vitro colony formation and migration of Pten^Δ/Δ^ HSCs and drives MPN development. Pharmacological inhibition of Rac by NSC-23766 prolonged the survival of the mice and reduced splenomegaly [120].

In KITD16V/KITD814V AML cells, both Rac1 and Rac2 are essential for ligand-independent growth of these leukemic cells. When primary BM cells derived from mice lacking the expression of Rac1, Rac2, or both were transfected with KITD814V, ligand-independent growth was significantly reduced in all three conditions compared to the control. The strongest effect was observed in the situation of double deletion (~75% reduction) whereas single deletions of Rac1 and Rac2 resulted in a reduction of about 10%–15% and 50%, respectively. Reintroduction of active Rac1/Rac2 into these cells restores ligand-independent growth, underscoring the crucial role of Rac GTPases in the proliferation of these cells [57]. One year later, the same group reported involvement of Rac1 in another important leukemic pathway. As described above, KITD816V and FLT3ITD-bearing oncogenic cells possess constitutively active FAK [58]. Downstream of active FAK, a signaling pathway was identified involving Tiam1/Rac1/Pak1, which finally leads to STAT5 activation and nuclear translocation. In vitro or in vivo inhibition of any member of this pathway suppresses strikingly the constitutive growth of the leukemic cells and results in repression of STAT5 activation and translocation. This is accompanied by reduced c-myc and Bcl-xL expression levels. Furthermore, in a model using transplantation of a murine cell line (32D) transfected with FLT3ITD, additional silencing of Tiam1 or treatment with the FAK inhibitor, F-14, both delay MPN onset and increase the survival of the mice compared to control cells. Similarly, transplantation of murine PAK^−/−^ KITD814V primary BM cells display significantly delayed leukemogenesis and prolonged survival of the mice compared to a PAK proficient setting [57]. In HCL cells, Rac1 seems to be implicated in the proliferation process as well, as overexpression of its dominant-negative form in a human HCL cell line (BNBH-1) resulted in reduced proliferation rates and suppressed lamellipodia formation. However, diminished proliferation is also caused by overexpression of dominant-negative RhoA or Cdc42, suggesting an interplay with other RhoGTPases in this process [62].

In a murine CML model induced by p210-BCR-ABL-expression, Rac2 is clearly a key player of leukemogenesis [61]. Deletion of Rac2 alone, but not of Rac1, delayed leukemia onset and increased the survival of the mice significantly. Combined Rac1 and Rac2 loss further increased the delay, suggesting a synergistic effect of both isoforms, but with a clear dominance for Rac2. Nevertheless, Rac1/Rac2 double knockout mice develop CML-like myeloproliferative neoplasms among other tumors, which are most likely attributable to Rac3 activity. Hyperproliferation, a feature of p210-BCR-ABL transduced cells, is abrogated in a dose-dependent manner by pharmacological inhibition of Rac1 and Rac2. These findings are supported by the observation that only the depletion of Rac2, but not Rac1, results in growth inhibition and colony formation deficiencies of BCR-ABL-transduced CD34^+^ CB cells. These findings highlight a central role for Rac GTPases, with a particular role for Rac2, in p210-BCR-ABL-induced leukemogenesis and proliferation. Similarly, there is evidence that Rac2 via Vav3 involvement promotes cell cycle progression of p190-BCR-ABL^+^ B-ALL cells [59]. In addition, Rac3 appears to be implicated in p190-BCR-ABL-induced leukemogenesis, as genetic knockdown of Rac3 significantly increases the survival in the mouse model. Notably, this was only observed in the female cohort of p190-BCR-ABL Rac3^−/−^ transgenic mice [60].

## 5. Rac GTPases as Therapeutic Targets in Cancer

Despite the complex network of regulation (some of the most important pathways are summarized in Figure 1), Rac is an interesting therapeutic target due to its converging and central role as a signaling hub. Consequently, several attempts have been undertaken at developing antagonists inferring with Rac1 and its regulatory network. However, Rac1 and its primary GEFs and GAPs are not easily druggable targets and effective manipulation of the Rac1 network may not be a matter of inhibiting overall Rac activity, but of interfering with the right Rac-interaction at the right dose and therapeutic window.

### 5.1. Targeting Rac-GEF Interactions

In this context, a first main mechanism under investigation has been the disruption of the GTPase interaction with its specific GEF. The first small molecule identified was NSC-23766 [121]. It prevents the interaction of Rac1 with its GEFs, Tiam1 and Trio, by fitting into a surface groove of Rac1 and thereby further affecting the downstream activation of Rac1 effectors, such as PAK1. The treatment of CLL cells with this inhibitor leads to their reduced proliferation [52]. Also, p210-BCR-ABL CML cell proliferation could be reduced by NSC-23766 treatment [122]. In mantle cell lymphoma (MCL), concomitant NSC-23766 inhibitor treatment significantly increased the cytotoxic effect of Adriamycin [64].

Although, NSC-23766 was initially published as specific for Rac1, it has meanwhile been shown by different groups that NSC-23766 also targets Rac2, at least in p210-BCR-ABL^+^ HSPCs/CML cells [61,96]. Further, NSC-23766 displays critical off-target effects in mouse platelets, such as receptor downregulation, which pose a serious challenge to its further therapeutic development [123].

Distinct from NSC-23766, ZINC69391 and 1A-116 are two Rac1 inhibitors chemically related to each other, which were shown to reduce cell cycle progression, proliferation, and survival in AML cell lines [124]. ZINC69391 interferes with Rac1-Tiam1 binding and its analog, 1A-116, blocks the Rac1 interaction with P-Rex1. The small molecule compound, ITX3, inhibits Rac1-TrioN GEF binding, but the high mean inhibitory concentration (IC50) restricts clinical application [125].

Another inhibitor, namely EHop-016, was initially reported to be specific for Rac1 and Rac3 [126]. The mechanism of inhibition includes the binding of EHop-016 to key residues of Rac, thereby inhibiting Vav interaction. In SM and AML derived cells, where EHop-016 targets Rac1, Rac2, and Vav1 activation, this resulted in decreased effect of mutated KITD814V on cell growth [57]. However, the modest bioavailability and the high effective concentration need to be considered for further application [127]. Along this reasoning, the goal to develop a Rac inhibitor with improved activities recently resulted in the design of MBQ-167 [128]. Compared with EHop-016, MBQ-167 has a 10-fold higher potency to inhibit Rac, but is also a highly efficient inhibitor of Cdc42. Notably, in preclinical breast cancer models, MBQ-167 successfully antagonized mammary tumor growth and metastasis. Particularly considering its effects in the context of an oncogenic RAS–dependent cell line and its inhibition of downstream STAT3 phosphorylation [128], this drug is an interesting new candidate to be tested in the hematological context.

### 5.2. Targeting Rac-Nucleotide Interactions

Another strategy of targeting Rac is to inhibit nucleotide binding, and thus ultimately Rac downstream signalling. Only a few agents have been detected in this class, including EHT 1864, which inhibits Rac signalling via guanine nucleotide displacement [129]. In AML1-ETO9a cells, this inhibitor reduces homing capacity and induces apoptosis [47]. As NSC-23766, EHT 1864 revealed critical off-target effects in mouse platelets, such as platelet apoptosis, raising concerns about the safety of this drug [123].

Mitoxantrone (MTX) is an US food and drug administration (FDA) approved topoisomerase 2 inhibitor, which also interferes with GTP binding of Rac1, Cdc42, and RhoA. It inhibits filamentous actin reorganization and ultimately cell migration [130]. In children with a first relapse ALL, the 3-year overall survival was 45.2% upon idarubicin treatment, an established drug in relapsed ALL, versus 69.0% upon MTX treatment [131]. Because of the multi-inhibition of Rac1, RhoA, and Cdc42, MTX induces significant adverse off-target effects.

### 5.3. Targeting Rac Spatial Regulation

Since Rac GTPase localization to cell membranes is regulated by post-translational lipid modification, strategies of inhibiting those modifications are under investigation. For example, the geranylgeranyltransferase I inhibitor, GGTI-2418, prevents the addition of a geranylgeranyl group to proteins with a CAAX motif, and has shown an anticancer effect in preclinical and clinical trials [132]. In MM, GGTI-2418 reduced the percentage of MM tumors in the bone marrow and induces apoptosis [133]. Another class of inhibitors are statins, which suppress cholesterol production and inhibit HMG-CoA reductase to prevent prenylation of Rho GTPases. In further consequence, lipid modifications are decreasing and the nuclear Rac1 pool is degrading, although these drugs are rather nonselective. However, despite some early preclinical reports on reduced tumor growth of AML cell lines treated by statins [134,135], the therapeutic success thus far appears too limited, and more biological understanding and development is required.

### 5.4. Targeting Rac Downstream Effectors

Targeting Rac downstream effectors is another strategy to inhibit Rac signalling. IPA-3 is a direct and non-competitive PAK1 inhibitor, which forces PAK1 into a catalytically inactive conformation [136]. This agent induces cell death and decreases cell adhesion in human leukemic cell lines [137]. Several other PAK inhibitors have been identified, such as OSU-03012 [138] and FRAX597 [139], but all PAK inhibitors have yet to receive FDA approval.

Since Rac GTPases are hard to target due to their structure and limited binding pockets for small molecules, it is not an easy task to define new targeting approaches. Nevertheless, several other mechanisms and drugs are under investigation and there is continuous improvement of existing approaches as described above [128], which is important since Rac is upregulated in over 10% of cancer cases, remaining a valid potential target in tumorigenesis [140].

## 6. Final Remarks

Deregulation of Rac expression and activation plays a fundamental, but not yet fully understood, role in the development and progression of cancers. The current evidence linking Rac not only to metastasis and the development of resistance in solid tumors, but also to leukemic infiltration and progression of various hematologic malignancies highlights the need for a better understanding of this key integrator of microenvironmental signals in leukemia and lymphoma. Particularly, as the development of drugs targeting Rac or related signaling still lags behind other therapeutic developments, a better understanding of the comprehensive molecular contributions and networks will contribute to the opening of new avenues of insights and to better definitions of therapeutic windows and opportunities for single and combinatory approaches.

## Figures and Tables

**Figure 1 ijms-19-04041-f001:**
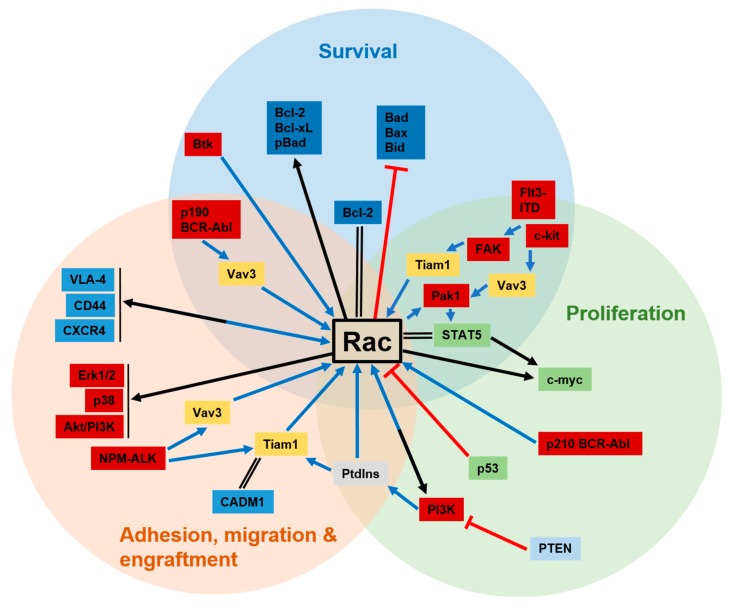
Schematic illustration of Rac signaling pathways contributing to adhesion, migration and engraftment, survival, and proliferation of hematological tumor cells. Depicted are upstream and downstream molecules and interaction partners of Rac GTPases that have been found in the different types of leukemia and lymphoma. Rac activating pathways are represented by blue arrows, direct or indirect activation downstream of Rac are represented by black arrows, direct interactions are marked by double black lines, and blocking processes are indicated by red bar-headed lines.

**Table 1 ijms-19-04041-t001:** Deregulations of Rac GTPases.

Malignancy	Model	Deregulation	Rac Isotype/GEF	Signaling Pathway	Reference
ALCL	Human: NPM-ALK^+^ cell lines	Constitutively active	Rac1	NPM-ALK → Vav3 → Rac1 → p38, Erk1/2, Akt	[54]
AML	Human: primary	Overexpression	Rac1		[55]
AML	Human: Primary pediatric MLL^+^	Overexpression	Rac2		[56]
AML	Human and mouse: KITD816V^+^ and FLT3ITD^+^ primary and cell lines	Constitutively active	Rac1	FAK → Tiam1 → Rac1 → Pak1 → STAT5 → c-myc, Bcl-xL	[57]
AML	Human and mouse: KITD816V^+^ primary and cell lines	Constitutively active	Rac1/Rac2	KITD816V → Vav1 → Rac1/Rac2 → Pak	[58]
B-ALL	Human and mouse: p190-BCR-ABL^+^ primary	Overexpression	Vav1, Vav3	Vav3 → Rac2 (Rac1, Cdc42, Rho)	[59]
B-ALL	Mouse: p190-BCR-ABL^+^ primary	Elevated active	Rac3		[60]
CML	Human and mouse: p210-BCR-ABL^+^ primary	Elevated active	Rac1, Rac2, Rac3	Rac1/Rac2 → Akt, JNK, p38, CrkL, STAT5	[61]
HCL	Human: cell line	Constitutively active	Rac1	PKCε → Src → Vav → Rac1 → p60Scr	[62,63]
MCL	Human: primary and cell lines	Overexpression	Rac1		[64]
MM	Human: primary	Overexpression, enhanced active, mutated (Rac1N92K)	Rac1		[53,65]

Depicted are the different types of Rac deregulations in hematological malignancies with, respectively, the model system used and the signaling pathway proposed.

**Table 2 ijms-19-04041-t002:** Contribution of Rac1 to migratory processes.

Malignancy	Model	Contribution	Rac Isotype	Signaling Pathway	Reference
ALCL	Human and mouse: ALK^+^ primary and cell lines	Migration, invasion, dissemination	Rac1	PtdIns5P-Tiam1 → Rac1 NPM-ALK → Vav3 → Rac1 → p38, Erk1/2, Akt	[54,81]
ALL	Human: cell lines	Actin protrusions, migration, engraftment	Rac1	CXCL12 → CXCR4-CD9 → Rac1	[82]
AML	Human: Primary Mouse: AML1-ETO9a (AE9a)	Migration, engraftment	Rac1	Rac1 → CD44, VLA-4, CXCR4	[47]
ATL	Human: cell lines	Adhesion lamellipodia formation,	Rac1	CADM1-Tiam1 → Rac1	[83]
MM	Mouse: primary and cell lines	Adhesion, firm arrest	Rac1	CXCL12 → CXCR4 → RhoA → ROCK → Rac1 → FAK-Src → VLA-4	[65,84]

Shown are the hematological malignancies and the specific contribution of Rac1 to migratory processes. The model systems used and the signaling pathways proposed are depicted.

**Table 3 ijms-19-04041-t003:** Contribution of Rac to leukemic/lymphoma cell survival and chemoresistance.

Malignancy	Model	Contribution	RAC Isotype/GEF	Signaling Pathway	References
ALCL	Mouse: ALK^+^ primary	Survival, proliferation, lymphomagenesis	Rac1		[89]
AML	Human: MLL^+^ primary and MA9^+^ cell lines	Survival	Rac1	BTK → Rac1	[90]
AML	Human and mouse: FLT3ITD^+^ primary and cell lines	Survival, proliferation, leukemogenesis	Rac1	FAK → Tiam1 → Rac1-Pak1-pSTAT5 → c-myc, Bcl-xL	[58]
AML	Human and mouse: KITD816V^+^ primary and cell lines	Survival, proliferation, leukemogenesis	Rac1/Rac2	KITD816V → Vav1 → Rac1/Rac2 → Pak → Bad	[57]
AML	Mouse: MA9^+^ primary	Survival	Rac2	Rac2 → Bcl-2, Bcl-xL	[91]
AML	Human: cell line	Chemoresistance	Rac1		[92]
B- and T- leukemia/lymphoma	Human: cell lines	Survival, chemoresistance	Rac1	Rac1-Bcl-2 → superoxide → pStat3 (Y705)	[93,94]
B-ALL	Human and mouse: p190-BCR-ABL^+^ primary	Survival, leukemogenesis	Vav3	Vav3 → Rac2 (Rac1, Cdc42, Rho)	[59]
Burkitt’s lymphoma	Human: cell line	Survival, proliferation, lymphomagenesis	Rac1	Rac1 → PKA → pBad	[95]
CLL	Human: primary	Chemoresistance, proliferation	Rac1	CD40L → Tiam1 → Rac1 → c-myc	[52]
CML	Human: primary CD34^+^ BCR-ABL^+^ CB cells and cell line	Survival, proliferation	Rac2		[96]

Depicted are the hematological malignancies and the specific contribution of Rac GTPases to survival and chemoresistance mechanisms therein. The model systems used and the signaling pathways proposed are shown.

**Table 4 ijms-19-04041-t004:** Contribution of Rac to leukemic/lymphoma cell proliferation.

Malignancy	Model	Contribution	Rac Isotype/GEF	Signaling Pathway	Reference
AML	Human and mouse: FLT3ITD^+^ and KITD816V^+^ primary and cell lines	Proliferation, survival, leukemogenesis	Rac1	FAK → Tiam1 → Rac1-Pak1-pSTAT5 → c-myc, Bcl-xL	[58]
AML	Human and mouse: KITD816V^+^ primary and cell lines	Proliferation, survival, leukemogenesis	Rac1/Rac2	KITD816V → Vav1 → Rac1/Rac2 → Pak → Bad	[57]
B- and T lymphoma	Human: cell lines	Proliferation, survival	Rac1	Rac1 → Pak1—Akt	[100]
B-ALL	Mouse: p190-BCR-ABL^+^ primary	Proliferation, survival, leukemogenesis	Vav3	Vav3 → Rac2 (Rac1, Cdc42, Rho)	[59]
CLL	Human: primary	Proliferation, chemoresistance	Rac1	CD40L → Tiam1 → Rac1 → c-myc	[52]
CLL	Human: Primary and cell line	Proliferation	Rac1/Rac2	Wnt5a → ROR1-ROR2 → ARGEF2, ARGEF6, DOCK2 → Rac1/Rac2	[110,111]
CML	Mouse: p210-BCR-ABL^+^ primary	Proliferation, survival	Rac2	Rac1/Rac2 → Akt, JNK, p38, CrkL, STAT5	[61]
CML	Human: primary CD34^+^ BCR-ABL^+^ CB cells and cell line	Proliferation, survival	Rac2		[96]
HCL	Human: cell line	Proliferation	Rac1	PKCε → Src → Vav → Rac1 → p60Scr	[62,63]

Depicted are the hematological malignancies and the specific contribution of Rac GTPases to proliferation processes therein. The model systems used and the signaling pathway proposed are shown.

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
