# Peer review of "Rac GTPases in Hematological Malignancies"

_ijms, 2018, doi:10.3390/ijms19124041_

Reviewer 1 Report

This review examines the role of Rac GTPases in haematological malignancies.

This review is well written, very detailed and updates knowledge about Rac in the haematological malignancies. However, there are several points that need to be addressed by the authors:

- A short paragraph on haematological tumors should appear in the introduction.

- What is very interesting in this review is the study of Rac in the haematological system. There have already been many reviews giving general information about Rac. It seems important to me to lighten Chapter 1 and refer to reviews for generalities and to focus on Rac in the haematological system. For example, instead of two rather long and general paragraphs on GEFs, it would be interesting to focus this section on Rac GEFs in haematological cells.

-Tables 2, 3 and 4 do not provide additional information compared to the text. It would really be appreciated to replace them with figures. These figures would show in which directions the regulation is carried out (activation or inhibition), which is not mentioned in the table, and would make it possible to identify mechanisms common to several haematological malignancies.

Minor comments:-line 119: I will remove “and cdc42”. Rac is the main regulator for lamellipodia formation. -line 134-136: As migration towards M-CSF-1 is directed migration, could you reformulate the sentence ?-lines 194-195; lines 205-208; lines 226-228: I wonder if it is relevant to talk about solid tumors again in Chapter 3: if the authors think it is relevant, could they add the driver mutation in melanoma (P29S)?-line 256. Could the authors remove “respectively” which refers to the references?-line 281: Could the authors add “similarly to Rac in normal” ?-line 315, the authors add to define what is NSC23766.-line 349: Could the authors specify what they mean by the term “normally” ?-line 458: Could the authors use “higher” instead of “increased” ?-line 470: Could the authors write “result in” instead of “resultin” ?-line 485: Could the authors write “de novo” in italics ?-line 497: Could the authors write “underscoring” instead of “undercoring” ?-line 564: Could the authors write “progression” instead of “prgression” ?  

Author Response

Point-by-point reply to Reviewer 1:

We would like to thank the reviewer for the insightful comments and thoroughly corrections and have incorporated the suggestions as outlined below. The modifications can be followed in the manuscript as they are marked in red.

Comment 1: “A short paragraph on haematological tumors should appear in the introduction.”

Response 1: Following this valuable suggestion, we have incorporated a new paragraph, which introduces hematological malignancies in the context of this review (lines 177-186).

 Comment 2: “What is very interesting in this review is the study of Rac in the haematological system. There have already been many reviews giving general information about Rac. It seems important to me to lighten Chapter 1 and refer to reviews for generalities and to focus on Rac in the haematological system. For example, instead of two rather long and general paragraphs on GEFs, it would be interesting to focus this section on Rac GEFs in haematological cells.”

Response 2: We fully agree with this comment and have shortened and lightened the respective section so that the focus is on Rac rather than on general aspects (lines 75-81).

Comment 3: “Tables 2, 3 and 4 do not provide additional information compared to the text. It would really be appreciated to replace them with figures. These figures would show in which directions the regulation is carried out (activation or inhibition), which is not mentioned in the table, and would make it possible to identify mechanisms common to several haematological malignancies.”

Response 3: We addressed this comment by submitting an additional figure (lines 549-556) while still keeping the tables. This way, the figure gives an overview of the regulation pathways, including the direction of regulatory interactions, as suggested by the reviewer. The tables deliver complementary details to the readers, allowing easy referral to the relevant citations. 

Minor comments: We are very grateful to this careful proof-reading and corrected the errors.

Reviewer 2 Report

The review entitled “Rac GTPases in hematological Malignancies” is very interesting. It is well written and very informative regarding the role of Rac GTPases in cancer and particular in hematologic malignancies.

It describes globally the upstream and downstream molecules participating in RAC pathway.

The only information that it will be probably interesting to be added is the roe of PI-3 kinase upstream of RAC GTPase pathway since there are important studies pointing this interaction..

It will be also interesting if the authors describe the effect of the new drug MBQ-167 in tumor cells.

Author Response

Point-by-point reply to Reviewer 2:

We thank the reviewer for the insightful comments and helpful suggestions. We have gladly incorporated the suggestions and the new paragraphs and sections are marked in red. In detail:

Comment 1: “The only information that it will be probably interesting to be added is the roe of PI-3 kinase upstream of RAC GTPase pathway since there are important studies pointing this interaction”

Response 1: Following this helpful suggestion, we added a respective paragraph addressing the PTEN-PI3K relationship and its association to the Rac pathway in lines 497-505. This information is also depicted in the newly designed Figure 1 (line 549ff).

Comment 2: “It will be also interesting if the authors describe the effect of the new drug MBQ-167 in tumor cells.”

Response 2: Indeed, we had overlooked interesting recent developments. We are grateful for this hint and have carefully revised and reordered the relevant section (line 579ff) with new information in lines 589-595 and 629-631, in accordance with the Reviewer’s suggestion.